# Optimization of Ecological Water Replenishment Scheme Based on the Interval Fuzzy Two-Stage Stochastic Programming Method: Boluo Lake National Nature Reserve, Jilin Province, China

**DOI:** 10.3390/ijerph19095218

**Published:** 2022-04-25

**Authors:** Hao Yang, Wei He, Yu Li

**Affiliations:** MOE Key Laboratory of Resources Environmental Systems Optimization, North China Electric Power University, Beijing 102206, China; 120212232080@ncepu.edu.cn (H.Y.); 120202232011@ncepu.edu.cn (W.H.)

**Keywords:** interval fuzzy two-stage stochastic programming (IFTSP) method, Boluo Lake National Nature Reserve, wetland, ecological water replenishment, decision-making risks

## Abstract

In this paper, a fuzzy mathematical programming method is introduced on the basis of the interval two-stage stochastic programming (ITSP) optimisation model for the wetland ecological water replenishment scheme in Boluo Lake National Nature Reserve. The minimum ecological water supply is taken as the objective function, and the lake bubble water diversion capacity, lake bubble water supply capacity, water diversion sequence, ecological service value, and minimum capacity of the wetland water supply are taken as constraints. The ecological water replenishment schemes of five lakes in the Boluo Lake National Nature Reserve are optimised at the levels of low flow years, normal flow years, and high flow years, and an optimised model for the wetland ecological water replenishment scheme in Boluo Lake National Nature Reserve based on the interval fuzzy two-stage stochastic programming (IFTSP) method is constructed. The model fully considers the waste of water resources and the protection of migratory bird habitat and makes rational allocation of water resources to make full use of flood resources. The IFTSP model proposed herein fully considers the fuzzy and uncertain characteristics of the planning area in the lake bubble area of Boluo Lake National Nature Reserve and improves the decision-making efficiency of decision-makers by providing technical support for smooth implementation of the ecological water replenishment project in nature reserves. The model can also be used as a theoretical guide for ecological recharge projects in other regions of the world.

## 1. Introduction

Water is one of the most precious natural resources, and the survival and development of all organisms is inseparably linked to the availability of water [1,2]. China ranks sixth in the world in terms of total water resources, and generally, water resources are relatively abundant in the country. However, owing to more precipitation in summer and autumn and less precipitation in winter and spring, the annual distribution of water resources is unbalanced. Meanwhile, there are considerable differences in precipitation levels among low flow years, normal flow years, and high flow years; these differences lead to uneven distribution of water resources between years [3,4,5]. Due to the annual and interannual precipitation imbalance, the spatial and temporal distributions of water resources in China are uneven, which can easily cause either floods or drought in several regions. Jilin province is located in the northern part of China, and it experiences a temperate monsoon climate; summer precipitation accounts for 60% of the annual precipitation, whereas winter precipitation accounts for less than 5%. Spatially, water resources are more abundant in the eastern mountain areas and less abundant in the river network in the western plains. The total amount of water resources is insufficient, and severe water shortages are experienced in the region. Water resources vary significantly across various regions in China, and the northern region, which is north of the Qinling–Huaihe River, accounts for 63.5% of the country’s total water resources, but the freshwater resources only account for 19.0% of the country’s total water resources. In particular, the per capita water resource in the Songliao River Basin is lower than the national average. The Songhua River Basin falls in the area with moderate water shortage, whereas the Liao River Basin falls in the area with severe water shortage, meaning that water shortage is prevalent in some parts of China [6,7,8]. In this light, it is especially important to allocate water resources scientifically and improve the utilisation rate of water resources [9,10,11].

Wetlands, also called the ‘kidneys’ of the Earth, are not only among the most important living environments in nature but also the most biodiverse regions, and they occupy an important and irreplaceable position in the ecosystem [12,13,14]. Changes in natural wetland environments can induce changes in soil, climate, hydrology, and other factors, and when these factors change, they can induce changes in the functions of wetland ecosystems to a greater or lesser extent [15,16]. This is especially true for the hydrological environment, which is the most strongly affected due to natural disturbances or human factors; consequently, biological habitats and amounts of water resources are reduced, and the survival of biological species becomes challenging. Moreover, the structure of the biological community may be affected and biodiversity may be destroyed, leading to changes in the structure of wetland ecosystems and destruction of their stability [17,18]. Therefore, it is extremely important to allocate water resources to wetlands in a scientific and rational manner to secure their ecological functions and stability [19].

Thus far, water resource allocation has been researched extensively: Huang et al. [20] combined a distributed hydrological model and an interval two-stage stochastic planning approach to rationally allocate water resources in the Kaidu–kangqi Basin and achieve the highest economic efficiency while satisfying the water demand of each sector. Li et al. [21] introduced interval parameters in two-stage stochastic planning to construct uncertain two-stage water allocation models for the planning and management of water resources with the aims of allocating water resources scientifically and rationally and reducing water wastage while maximising the net benefits of the system. Meng et al. [2] constructed an economic maximisation model for water recharge based on an interval two-stage stochastic planning approach to address the uncertainty of water consumption in the industrial, municipal, environmental, and agricultural sectors of the Yinma River Basin; in addition, they conducted a simulation to determine the highest economic benefits in each sector while fulfilling the water consumption requirements. However, these studies did not consider the fuzzy uncertainty of the ecological recharge region in the basin lake bubble area in water resources planning and management, which can easily lead to water resource wastage. Moreover, the simulation result interval of the model is too large, which affects the decision-making process in water resource management. Huang et al. [22] adopted the interval two-stage stochastic programming method to replenish water in the Boluo Lake National Nature Reserve while minimizing ecological replenishment. Although this simulation model was able to reduce ecological replenishment, it did not fully consider the effects of regional water supply uncertainty on water replenishment, resulting in water resource wastage. Moreover, the simulation scope of the model was broad, which is inconducive from the viewpoint of helping decision-makers to implement lake bubble recharge in the Boluo Lake National Nature Reserve.

The core idea of fuzzy mathematical programming is to represent fuzzy uncertainty in the form of fuzzy numbers, transform it into a general linear programming problem, and solve it to obtain the optimal solution under loose conditions [23]. Fuzzy mathematical programming mainly considers the maximum and minimum possible parameter values, and it requires a small amount of data. For these reasons, it is widely used in the planning and management of water resources. To address the problems of water resource management under uncertainty and the allocation of water demand by decision-makers in municipalities, industrial units, and agricultural sectors Maqsood et al. [23] proposed an interval fuzzy two-stage stochastic planning approach to account for the uncertainty of future water demand in each sector. In this model, the uncertainty was represented as a probability distribution with discrete intervals to rationalise the water allocation problem. Lu et al. [24] targeted the problem of water resource management under uncertain conditions by assuming a water resources management problem, allocating water to three users from a hypothetical reservoir, and constructing three models, namely two-stage stochastic programming (TSP), interval two-stage stochastic programming (ITSP), and interval fuzzy two-stage stochastic programming (IFTSP). A comparison of the simulation results obtained using these three models revealed that the IFTSP model can generate more system benefits and help one to manage water resources more scientifically. Li et al. [25] developed an interval fuzzy two-stage model to express the uncertainty of carbon dioxide emissions from industrial systems in the form of a probability distribution and discrete intervals with the aim of optimizing the carbon dioxide emissions from each source while fulfilling the carbon dioxide emissions requirements and maximising system benefits.

In summary, herein, we introduce a fuzzy mathematical planning method based on the ecological water replenishment model of the Boluo Lake National Nature Reserve, which was developed starting from the interval two-stage optimisation method proposed by Huang et al. [22]. The functional area of the lake bubble recharge area is blurred, and an ecological water replenishment model of the Boluo Lake National Nature Reserve by means of interval fuzzy two-stage stochastic programming (IFTSP) has been constructed to allocate water resources scientifically and rationally, avoid water resource wastage due to unreasonable water resource allocation, improve the decision-making efficiency of decision-makers, and provide a highly reasonable water replenishment scheme for decision-makers.

## 2. Study Area

Boluo Lake National Nature Reserve is located in Changchun City, west of Nongan County. Boluo Lake, which is located in the west of Boluo Lake National Nature Reserve, is the largest bubble in Changchun City. Moreover, it is the third largest bubble pond in Jilin Province and the only large-scale natural wetland in Changchun City. It is called the ‘natural kidney’ of Changchun City. The Boluo Lake National Nature Reserve covers an area of 24,915 hectares, of which 8600 hectares are wetlands during the period with abundant water, 6100 hectares during the flat flow water period, and 2857 hectares during the dry period [22]. A schematic diagram of the Boluo Lake National Nature Reserve is presented in Figure 1. Boluo Lake wetland has an important position, mainly because of its strong ability to regulate the ecological environment, especially the local weather. Meanwhile, the lakes in this reserve are important habitats and breeding places for aquatic organisms and birds [26]. The transpiration of Boluo Lake wetlands can adjust local humidity, increase local rainfall, and facilitate flood regulation and storage, thus playing a vital role in flood control and disaster reduction in the basin. For these reasons, the Boluo Lake wetland is an important ecological protection barrier for the natural ecology west of Changchun City. Given that Boluo Lake is a closed water area, there is a lack of large-scale water diversion projects in the territory. The lake and river are not connected, and the region is dry throughout the year. Water storage in the region has been decreasing year on year, and the lake surface is gradually shrinking, resulting in vegetation withering, bird migration, environmental deterioration, and loss of ecological and environmental benefits. Meanwhile, the decrease in precipitation and persistently high temperatures in recent years have led to droughts and other disasters. Moreover, excessive man-made reclamation, construction of ponds and river dams, and other water conservancy projects have destroyed wetland vegetation, resulting in soil erosion and other natural disasters. In addition, these projects have led to water resource imbalance within Boluo Lake National Nature Reserve, which hinders fulfilment of the conditions associated with water conservation, causing severe damage to the ecological environment of Boluo Lake [22]. Therefore, it is necessary to replenish the wetlands with water; allocate water resources rationally; stall the ecological deterioration of the Boluo Lake National Nature Reserve and the surrounding ecological environment; restore the wetland area and its ecosystem functions; reinvigorate Boluo Lake, which has been dry for many years; increase precipitation in the Boluo Lake National Nature Reserve and its surrounding areas; and reproduce the natural scenery of wetlands with grass fertiliser, water, and birds and flowers.

## 3. Model Construction

### 3.1. Overview of Interval Fuzzy Two-Stage Stochastic Programming Methods

The fuzzy mathematical programming method mainly fuzzified uncertain parameters and represents variables in the form of fuzzy random variables. A fuzzy problem is transformed into a general linear programming problem and solved to obtain the optimal model solution [27,28,29]. The expression of the interval fuzzy two-stage stochastic programming method after coupling the fuzzy mathematical method and the interval two-stage stochastic programming method is as follows [30,31,32].
(1)$$\max\lambda^{\pm }$$

Constraints:
(2)$$C^{\pm }X^{\pm }-\sum_{h=1}^{k}p_{h}q({y_{h}}^{\pm },{w_{h}}^{\pm })\geq f^{+}-(1-\lambda^{\pm })(f^{+}-f^{-})$$
(3)$$A^{\pm }X^{\pm }\leq b^{-}+(1-\lambda^{\pm })(b^{+}-b^{-})$$
(4)$$X^{\pm }\geq 0$$
(5)$$0\leq \lambda^{\pm }\leq 1$$
where $\lambda^{\pm }$ represents the fuzzy membership interval of the model; $A^{\pm }$ is an *m* × *n* matrix; $b^{\pm }$ is an m-dimensional vector; $C^{\pm }$ is an n-dimensional constant vector, $p_{h}$ represents the probability level of a random event; $X^{\pm }$ represents the first stage, which is the decision variable in the absence of random events; $y_{h}^{\pm }$ represents the second stage, which is the decision variable after the random event; $\omega_{h}^{\pm }$ is the random variable in the model; and $f^{+},f^{-}$ represent the upper and lower limits of the objective function of the interval two-stage stochastic programming model, respectively. By using an interactive algorithm, the model is divided into two sub-models considering the upper and lower bounds to solve the problem. The solutions are $\lambda_{opt}^{\pm }=[\lambda_{opt}^{-},\lambda_{opt}^{+}]$ and $X_{opt}^{\pm }=[X_{opt}^{-},X_{opt}^{+}]$, and the optimised value is $f_{opt}{}^{\pm }=C^{\pm }X^{\pm }-\sum_{h=1}^{k}p_{h}q\left(y_{h}{}^{\pm },\omega_{h}{}^{\pm }\right)$.

### 3.2. Construction of Optimised Ecological Water Supply Model for Boluo Lake Wetland Based on Interval Fuzzy Two-Stage Stochastic Programming Method

In the process of wetland water replenishment, the state and local governments are often uncertain about the planned water supply scope of lakes and wetlands, whereas the ITSP model does not take these factors into account, and the fuzzy uncertainty of these factors makes it difficult to effectively implement wetland water supply schemes and affect the recharge efficiency of wetland recharge projects. If only the minimum value of the lake bubble area is available, it is not conducive to restoring the ecosystem. If the maximum area of the lake bubble is to be restored, relatively more water is required, which may lead to water wastage due to insufficient scientific allocation [33]. Therefore, the uncertain factors encountered in the process of ecological water replenishment can be effectively addressed by fuzzifying the lake bubble recharge area and blurring the linear programming constraints and objective function to arrive at the optimal solution under relaxed conditions [28,34].

In this paper, a fuzzy method is introduced on the basis of the interval two-stage stochastic programming method. Moreover, an optimised model of ecological water replenishment of the Boluo Lake wetlands is developed using the interval fuzzy two-stage stochastic programming method with functional area and lake bubble area as the fuzzy variables. Moreover, minimum ecological replenishment is set as the optimisation objective by fully considering the uncertainty of the fuzzy variables such as lake bubble ecosystem planning, water surface area, and wetland area. Considering lake bubble diversion and water replenishment as constraints, optimal allocation of different types of water resources (local incoming water, normal water supply, and flood resources) is performed to minimise ecological replenishment and restore the ecosystem functions of the Boluo Lake wetlands.

The objective function of the IFTSP model constructed in this paper is as follows:(6)$$\max\lambda^{\pm }$$
where $\lambda^{\pm }$ represents the fuzzy membership interval of the model.

Constraints:

(1) Constraints on minimisation of ecological replenishment
(7)$$\sum_{i=1}^{5}EAT_{i}^{\pm }\cdot QT_{i}^{\pm }-\sum_{i=1}^{5}\sum_{j=1}^{4}\sum_{h=1}^{3}p_{h}\cdot DEA_{ijh}^{\pm }\cdot QLT_{ih}^{\pm }\leq f^{-}+\left(1-\lambda^{\pm }\right)\cdot \left(f^{+}-f^{-}\right)$$
where $f^{+}$ and $f^{-}$ represent the upper and lower limits of ecological water supply optimised using the interval two-stage stochastic programming model; i = 1 to 5 represent the Toudaogang Reservoir, Boluo Lake, Mobopao, Yuanbaowapao, and Aobaotupao, respectively, in the nature reserve; j = 1 to 4 represent the four different ecosystem types, namely, fish ponds, crab ponds, reed wetlands, and marsh wetlands, respectively; h = 1 to 3 represent the flood amount levels in the low flow years, normal flow years, and high flow years, respectively; $p_{h}$ denotes the scenario probability; $EAT_{i}^{\pm }$ represents the water replenishment area recommended by the project (hm^2^); $QT_{i}^{\pm }$ represents the total amount of water replenishment for the lake and pond (t/hm^2^); $DEA_{ijh}^{\pm }$ represents the amount of area adjustment for the lake and pond i in the ecosystem j under scenario h (hm^2^); and $QLT_{ih}^{\pm }$ represents the total water transport loss for the lake and pond i under scenario h.

(2) Constraints on water diversion and amount of supplementation [22]
(8)$$\left(AP_{i}^{\pm }-DAP_{ih}^{\pm }\right)\cdot QWR_{i}^{\pm }\leq QAP_{ih}^{\pm },\forall i,h$$
(9)$$\sum_{j=3}^{4}\left(EA_{ij}^{\pm }-DEA_{ijh}^{\pm }\right)\cdot QP_{ij}^{\pm }\leq QAW_{ih}^{\pm },\forall i,h$$
where $AP_{i}^{\pm }$ represents the water surface area of the lake and pond (hm^2^); $DAP_{ih}^{\pm }$ represents the reduction in the water surface area of lake and pond i under scenario h (hm^2^); $QWR_{i}^{\pm }$ represents the unit water demand quota of the water surface area of lake and pond i (m^3^/hm^2^); $QAP_{ih}^{\pm }$ represents the water supply amount of lake and pond i under scenario h (m^3^); $QP_{ij}^{\pm }$ represents the unit water demand quota of lake and pond i in ecosystem j (m^3^/hm^2^), and $QAW_{ih}^{\pm }$ represents the amount of water replenishment in the wetland of lake and pond i in scenario h (m^3^).

(3) Minimum water constraint
(10)$$QTF_{ih}^{\pm }\geq \sum_{j=3}^{4}\left(LOA_{ij}^{-}+\left(1-\lambda^{\pm }\right)\left(LOA_{ij}^{+}-LOA_{ij}^{-}\right)\right)\cdot QP_{ij}^{\pm }$$
where $LOA_{ij}^{\pm }$ is a fuzzy variable that represents the lower limit of the of lake and pond i (hm^2^).

(4) Constraints on water supply capacity [22]
(11)$$QAI_{ih}^{\pm }\leq QI_{i}^{\pm },\forall i,h$$
(12)$$QAN_{ih}^{\pm }\leq QN_{i}^{\pm },\forall i,h$$
where $QAI_{ih}^{\pm }$ represents the local amount of water in lake and pond i under scenario h (m^3^); $QI_{i}^{\pm }$ represents the local amount of water in the lake and pond (m^3^); $QAN_{ih}^{\pm }$ represents the normal supplement of lake and pond i under scenario h (m^3^); and $QN_{i}^{\pm }$ represents the normal supplement of the lake and pond (m^3^).

(5) Constraints on functional area
(13)$$AP_{i}^{\pm }-DAP_{ih}^{\pm }\geq 0,\forall i,h$$
(14)$$EA_{ij}^{\pm }-DEA_{ijh}^{\pm }\geq 0,\forall i,h$$
(15)$$\sum_{j=1}^{2}EA_{ij}^{\pm }-DEA_{ijh}^{\pm }\leq AP_{i}^{\pm }-DAP_{ih}^{\pm },\forall i,h$$
(16)$$\left(AP_{i}^{\pm }-DAP_{ih}^{\pm }\right)+\sum_{j=3}^{4}\left(EA_{ij}^{\pm }-DEA_{ijh}^{\pm }\right)\leq PLA_{ij}^{-}+\left(1-\lambda^{\pm }\right)\cdot \left(PLA_{ij}^{+}-PLA_{ij}^{-}\right),\forall i,h$$
where $PLA_{ij}^{\pm }$ is a fuzzy variable that represents the upper limit of the area of lake and pond i (hm^2^);

(6) Constraints on diversion and supplementation of water amount [22]
(17)$$\sum_{i=1}^{5}\left[\sum_{j=1}^{4}\left(EA_{ij}^{\pm }-DEA_{ijh}^{\pm }\right)\cdot QP_{ij}^{\pm }-QAI_{ih}^{\pm }-QAN_{ih}^{\pm }\right]\leq QTF_{ih}^{\pm },\forall h$$
where $QTF_{ih}^{\pm }$ represents the amount of flood resources in lake and pond i under scenario h (m^3^).

(7) Constraints on water diversion and supplementation order
(18)$$QAW_{ih}^{\pm }=\left\{ \begin{array}{l} \left(QAI_{ih}^{\pm }-QLI_{ih}^{\pm }\right)+\left(QAN_{ih}^{\pm }-QLN_{ih}^{\pm }\right)+\left(QAF_{ih}^{\pm }-QLF_{ih}^{\pm }\right)-QAP_{ih}^{\pm }\\ ifAP_{i}^{\pm }-DAP_{ih}^{\pm }\geq AP_{i\min}^{+}-\left(1-\lambda^{\pm }\right)\cdot \left(AP_{i\min}^{+}-AP_{i\min}^{-}\right)\\ 0,ifAP_{i}^{\pm }-DAP_{ih}^{\pm }\leq AP_{i\min}^{+}-\left(1-\lambda^{\pm }\right)\cdot \left(AP_{i\min}^{+}-AP_{i\min}^{-}\right) \end{array}\right.,\forall i,h$$
(19)$$\left(EA_{i4}^{\pm }-DEA_{i4h}^{\pm }\right)\cdot QP_{i4}^{\pm }=\left\{ \begin{array}{l} QAW_{ih}^{\pm }-\left(EA_{i3}^{\pm }-DEA_{i3h}^{\pm }\right)\cdot QP_{i3}^{\pm }\\ ifEA_{i3}^{\pm }-DEA_{i3h}^{\pm }\geq EA_{i3\min}^{+}-\left(1-\lambda^{\pm }\right)\cdot \left(EA_{i3\min}^{+}-EA_{i3\min}^{-}\right)\\ 0,ifEA_{i3}^{\pm }-DEA_{i3h}^{\pm }\leq EA_{i3\min}^{+}-\left(1-\lambda^{\pm }\right)\cdot \left(EA_{i3\min}^{+}-EA_{i3\min}^{-}\right) \end{array}\right.,\forall i,h$$
where $QLI_{ih}^{\pm }$ represents the loss of local water transport in lake and pond i under scenario h (m^3^); $QLN_{ih}^{\pm }$ represents the loss of water transport considering the normal supplementation of lake and pond i under scenario h (m^3^); $QLF_{ih}^{\pm }$ represents the loss of water transport considering flood diversion and supplementation in lake and pond i under scenario h (m^3^).

(8) Constraints on ecological value [22]
(20)$$TEBE_{kh}^{\pm }=\sum_{i=1}^{5}\sum_{j=1}^{4}EBW_{jk}^{\pm }\cdot \left(EA_{ij}^{\pm }-DEA_{ijh}^{\pm }\right)$$
(21)$$TEBE_{kh}^{\pm }\geq TEB_{k}^{\pm }$$
where k is the ecosystem service function of the reserve; $TEBE_{kh}^{\pm }$ represents the total ecological value of the indicator system under scenario h (10^4^CNY); $EBW_{jk}^{\pm }$ represents the ecological value compensation of ecosystem j (10^4^CNY); and $TEB_{k}^{\pm }$ represents the total ecological value of the index system of the project scheme (10^4^CNY).

(9) Nonnegative constraint:(22)$$QAI_{ih}^{\pm },QAN_{ih}^{\pm },QAF_{ih}^{\pm },QAW_{ih}^{\pm },QAP_{ih}^{\pm }\geq 0$$

In the optimisation scheme, the quantity of water replenishment for each lake and pond is nonnegative.

## 4. Results and Discussion

Herein, we use the IFTSP method to solve the ecological water replenishment model of the Boluo Lake National Nature Reserve in the Lingo18.0 software (LINDO, Chicago, IL, USA) environment. With the objective of minimum ecological replenishment, the ecological replenishment model based on the interval two-stage stochastic programming method constructed by Huang et al. [22] was fuzzified, while the IFTSP model constructed in this paper fully considered the fuzzy uncertainty of the lake bubble and wetland area in the Boluo Lake National Nature Reserve, effectively avoiding the waste of water resources. The fuzzy membership interval of the IFTSP model is [0.32, 0.64].

### 4.1. Analysis of the Change in Ecological Replenishment Water Quantity Used for Boluo Lake Wetland Water Replenishment Project Based on IFTSP Model

#### 4.1.1. Ecological Replenishment Water Configuration Scheme for Boluo Lake Wetland Water Replenishment Project Based on the IFTSP Method

The optimal allocation of ecological replenishment water in the Boluo Lake National Nature Reserve, as simulated using the IFTSP model, is summarised in Table 1.

Compared to the recommended project scheme, after model simulation and optimisation, the upper and lower limits of total ecological replenishment of the Toudaogang Reservoir in the Boluo Lake National Nature Reserve changed [−13.67%, 2.46%], and the lower limit of total ecological replenishment was significantly lower than the recommended value. By contrast, the upper limit increased to a value higher than the recommended value in the project. The upper and lower limits of total ecological replenishment of Boluo Lake changed [−53.04%, 6.71%], and the lower limit of total ecological replenishment decreased significantly, while the upper limit, too, decreased. Compared to the recommended values in the project scheme, the upper and lower limits of total ecological replenishment of Mobopao changed [−21.45%, 16.30%], and both the lower and upper limits of total replenishment decreased significantly. Compared with the recommended values in the project scheme, the upper and lower limits of total ecological replenishment of Yuanbaowapao changed [−100.00%, −74.58%], and both the upper and lower limits of total ecological replenishment decreased significantly. Compared to the recommended values in the project scheme, the upper and lower limits of total ecological replenishment of Aobaotupao changed [−10.07%, 48.42%], where the lower limit of total ecological replenishment decreased significantly, whereas the upper limit increased significantly. A comprehensive analysis of the above results indicates that although the amount of ecological recharge altered in the five lake bubbles varied. The lower limits of ecological recharge decreased in all cases, and the amounts of decrease ranged from 2.46% to 48.42%. Meanwhile, both the upper and lower limits decreased in the case of the Boluo Lake, Yuanbaowapao, and Mobopao, and the reductions were significant (Figure 2). Overall, the IFTSP model was able to ensure that decision-makers can adjust the ecological replenishment of each lake bubble according to the actual situation when allocating water resources to the Boluo Lake National Nature Reserve, thus optimizing the allocation of water resources and making full and rational use of water resources to reduce the amount of replenishment.

#### 4.1.2. Comparative Analysis of Ecological Recharge Configuration Options for Boluo Lake Wetland Recharge Project by Using IFTSP and ITSP Models

The optimised simulation results obtained using the ITSP and IFTSP models, which are summarised and shown in Table 2 and Figure 3, respectively, indicate that the total ecological water supply to the Toudaogang Reservoir after IFTSP model optimisation changed by [0.00, −25.45] × 10^4^ m^3^ compared to that after ITSP model optimisation. The upper limit of the total ecological water supply in this area decreased, and the interval range was shortened by 19.50%. Compared to the ITSP model, the total ecological replenishment of the Boluo Lake with the IFTSP model changed by [786.49, −1541.65] × 10^4^ m^3^, and the width of the interval was shortened from 5778.71 × 10^4^ m^3^ to 2450.57 × 10^4^ m^3^, a significant shortening of the decision range, with a 57.59% reduction in the decision range. Compared to the ITSP model, total ecological water replenishment of the Mobopao with the IFTSP model changed by [562.21, 55.19] × 10^4^ m^3^, and the decision width was shortened from 657.00 × 10^4^ m^3^ to 39.60 × 10^4^ m^3^, with a maximum reduction of 93.97% in the width of this lake bubble interval. Compared to the ITSP model, the total ecological water supply of Aobaotupao with the IFTSP model changed by [855.26, 0.00] × 10^4^ m^3^, and the decision range was significantly shortened from the original 1563.89 × 10^4^ m^3^ to 708.63 × 10^4^ m^3^, with a 54.69% reduction in the width of the interval. Overall, the IFTSP model exhibited an overall water recharge variation of [3203.96, −1622.29] × 10^4^ m^3^ compared to the ITSP model, and the total recharge decision range was significantly shortened from 8269.04 × 10^4^ m^3^ to 3442.79 × 10^4^ m^3^, with an interval range shortening ratio of 58.17%. This shows that the difference value between the upper and lower limits of the original model optimisation solution is obvious, which leads to an excessive decision range and tends to make decision-makers waste water resources and consume more decision time when carrying out ecological replenishment of the Boluo Lake National Nature Reserve. After the IFTSP model constructed in this paper is optimised, the upper limit of the interval is significantly reduced, effectively avoiding water wastage, the lower limit of the interval is significantly increased, and the width of the interval is significantly shortened. In summary, the IFTSP model helps to reduce the risk of decision-making, reduce the scope of decision-making and improve the efficiency of decision-making, while improving the construction efficiency of wetland recharge projects to shorten the construction period and provide decision-makers with a more scientific and reasonable ecological recharge plan.

### 4.2. Analysis of Water Diversion Variation in Boluo Lake Wetland Reserve Recharge Project Based on IFTSP Model

#### 4.2.1. Flood Diversion Water Allocation Scheme for Water Replenishment Project in Boluo Lake Wetland Reserve Based on IFTSP Model

Flood resources are one of the most important freshwater resources, and the full and rational use of flood resources can not only solve the problem of ecological water replenishment of Boluo Lake but also help to avoid disasters and maintain the stability of the Boluo Lake ecosystem. The IFTSP model constructed herein induces the flood volume configuration summarised in Table 3.

As can be inferred from Table 3, the flood resources are not used in the Toudaogang reservoir area in the recommended project scheme. Moreover, the flood resources are not used in the three water years in the Boluo Lake area considering the constraints of the water diversion sequence and the water supply capacity in the IFTSP optimisation scheme. With increasing precipitation in the low flow, normal flow, and high flow years, the amount of flood water diverted by each lake bubble in the Boluo Lake National Nature Reserve gradually decreased, and the upper and lower limits of flood water diverted by the Toudaogang Reservoir decreased from [270.36, 335.63] × 10^4^ m^3^ to [84.54, 177.12] × 10^4^ m^3^, respectively, indicating that the amount of flood water diverted in the low flow year was the highest. The upper and lower limits of the Mobopao flood diversion were [269.40, 307.21] × 10^4^ m^3^, [269.40, 478.80] × 10^4^ m^3^, and [12.60, 12.60] × 10^4^ m^3^, where the upper limit of annual flood diversion increased under the constraints of the water diversion sequence and flood resources. In Yuanbaowapao, as the annual precipitation changed, the upper and lower limits of flood diversion decreased from [206.55, 289.80] × 10^4^ m^3^ to [56.70, 289.80] × 10^4^ m^3^ and then to [56.70, 56.70] × 10^4^ m^3^. The upper and lower limits of flood diversion under different scenarios in Aobaotupao were [227.37, 556.39] × 10^4^ m^3^, [227.37, 556.39] × 10^4^ m^3^, and [227.37, 556.39] × 10^4^ m^3^. Overall, as the annual precipitation over the three flow years increased after optimisation, the upper and lower limits of flood diversion decreased from [973.68, 1489.03] × 10^4^ m^3^ to [638.01, 1502.11] × 10^4^ m^3^ and then to [381.21, 802.81] ×10^4^ m^3^. The amount of flood diversion not only meets the water supply needs of the wetlands but also conserves flood resources. Based on the above analysis, the flood diversion volume of each lake is different under different scenarios, but the overall flood diversion volume can comply with the value recommended in the project scheme, and the upper limit of the flood diversion volume is significantly higher than the value recommended in the project scheme. The amount of flood diversion in the three scenarios is different, mainly because of greater precipitation and adequate water resources in the rainy year, which means that the ecological replenishment can be reduced to some extent. Flood resources are often stored in reservoirs during the flood season in a rainy year and converted into available water resources. Therefore, flood resources can be allocated reasonably according to precipitation and ecological replenishment in different scenarios. The interval fuzzy two-stage model optimisation simulation results obtained herein fulfil the requirements outlined in the engineering recommendation scheme. Moreover, the model simulation interval represents the upper and lower limits of simultaneous indentation, and it significantly shortens the decision interval for decision-makers to allocate flood resources, which can effectively improve the decision-making efficiency.

#### 4.2.2. Comparative Analysis of ITSP and IFTSP Models for Diversion of Water from Boluo Lake Wetland Reserve Recharge Project

According to Table 4, after optimisation using the IFTSP model, the amount of flood diversion in the three flow years increased by 227.37 × 10^4^ m^3^ compared to the lower limit of the ITSP model; the upper limit decreased by 354.31 × 10^4^ m^3^, and the interval width was shortened from 910.70 × 10^4^ m^3^ to 329.02 × 10^4^ m^3^, a significant reduction in the decision range and a 63.87% reduction in the interval width. Compared to the ITSP model, the upper and lower limits of the flood diversion volume of Toudaogang Reservoir changed by [270.60, 0.00] × 10^4^ m^3^, respectively, and the decision range was significantly shortened from 335.63 × 10^4^ m^3^ to 65.27 × 10^4^ m^3^, with an interval width reduction ratio of 80.55%. Compared to the ITSP model, the upper and lower limits of the Toudaogang Reservoir changed by [84.54, −158.51] × 10^4^ m^3^, and the interval width was shortened from 335.63 × 10^4^ m^3^ to 92.58 × 10^4^ m^3^, a significant reduction in the decision range and a 72.71% reduction in the interval width. Compared to the ITSP model, the upper and lower limits of the flood diversion volume in the low flow year changed by [269.40, −122.89] × 10^4^ m^3^ in Mobopao, and the width of the interval was shortened from 431.10 × 10^4^ m^3^ to 37.81 × 10^4^ m^3^, with a maximum reduction in the decision range of 91.23%. Compared to the ITSP model, the upper and lower limits of the flood diversion volume in the normal water year changed by [269.40, 0.00] × 10^4^ m^3^, and the width of the interval was shortened from 431.10 × 10^4^ m^3^ to 209.40 × 10^4^ m^3^, a significant reduction in the decision range and a 56.27% reduction in the width of the interval. Compared to the ITSP model, the upper and lower limits of the flood diversion volume in the low flow year changed by [206.55, 0.00] × 10^4^ m^3^ in Yuanbaowapao, and the decision range was significantly shortened from 289.80 × 10^4^ m^3^ to 83.25 × 10^4^ m^3^, with a 71.27% reduction in the width of the interval. Compared to the ITSP model, the upper and lower limits of the flood diversion volume in the normal flow year changed by [56.70, 0.00] × 10^4^ m^3^, and the interval width decreased by 19.56%. Generally, the total flood diversion volume of each lake in the low flow water year changed by [973.68, −478.20] × 10^4^ m^3^ compared to the value obtained using the ITSP model, and the interval width decreased by 73.80%. The total flood diversion volume in the normal flow year changed by [638.01, −512.82] × 10^4^ m^3^, and the interval width decreased by 57.12%. Compared to the value obtained using the ITSP model, the total flood diversion volume in the high flow year changed by [381.21, −512.82] × 10^4^ m^3^, and the interval width decreased by 67.95%. After optimisation of the IFTSP model, the range of flood diversions in the three water years has been significantly shortened, and the upper limit of the simulation results has been significantly reduced, effectively avoiding the waste of flood resources, while the lower limit of the interval has been significantly increased, thus ensuring the full use of flood resources and shortening the decision-making range. In summary, the process was optimized by the interval fuzzy two-stage model to improve the decision-making efficiency of decision makers and help decision-makers to reduce the amount of flood diversion according to local conditions to meet the water demand in other areas. This helps to ensure that flood resources are fully utilised, and wastage of flood resources is avoided.

### 4.3. Analysis of Ecological Water Replenishment Variation in Boluo Lake Wetland Water Replenishment Project Based on IFTSP Method

#### 4.3.1. Functional Area Allocation Scheme for Water Replenishment Project in Boluo Lake Wetland Reserve Based on IFTSP Model

Wetland water replenishment is one of the most effective methods for recovering wetland areas and restoring their functions [35]. After optimisation using the IFTSP model proposed herein, changes in the size of each functional area in the reserve are indicated in Table 5.

According to Table 5, the reed wetland restoration area was the largest area after optimisation of the Toudaogang Reservoir by using the IFTSP model, and its size was [0.00, 0.96] × 10^4^ hm^2^. Although the lower limit was reduced by 0.06 × 10^4^ hm^2^ compared to the functional area in the current project, the upper limit increased by 700.00%. Compared to the functional area in the project, the lower limit of the fish pond area decreased, but the upper limit increased by 1340.00%. The lower limit of the marsh wetland decreased by 0.02 × 10^4^ hm^2^, whereas the upper limit increased by 725.00%. After simulation and optimisation of the IFTSP model in Boluo Lake, the recovery area of fish ponds was the largest at [6.62, 6.96] × 10^4^ hm^2^, which represents an increase of [859.42%, 435.38%] compared to the current functional area in the project, and the upper and lower limits of the other three functional areas increased considerably by [0.00%, 100.00%], [1303.85%, 700.00%] and [1376.92%, 779.17%] compared to the corresponding current functional areas in the project. After simulation and optimisation of the Mobopao by using the IFTSP model, the area of the crab pond did not change. The upper and lower limits of the fish pond area increased by [1000.00%, 700.00%], and the lower limits of the reed wetland and swamp wetland area decreased, but the upper limits increased by 700.00%. After simulation and optimisation of the Yuanbaowapao by using the IFTSP model, the upper and lower limits of marsh wetlands increased greatly by [1500.00%, 1000.00%], respectively. The upper and lower limits of the fish pond area remained unchanged. The lower limit of the crab pond area does not change, but upper limit increased by 100.00%. Compared to the current functional area in the project, the lower limit of the functional area of reed wetland decreased, but the upper limit increased by 900.00%. After simulation and optimisation using the IFTSP model, the upper and lower limits of the fish pond area did not change relative to the current functional area in the project in case of the Aobaotupao. However, the lower limits of crab ponds, reed wetlands, and swamp wetlands decreased, and the upper limits of the three functional areas increased greatly by 478.39%, 700.00% and 711.11%, respectively. The overall functional areas of the optimised regions increased greatly, indicating that the IFTSP model has a significant effect on restoring the functional areas in wetlands, thus ensuring the functional diversity of each region.

#### 4.3.2. Comparative Analysis of Changes in Functional Areas in Boluo Lake Wetland Reserve Water Replenishment Project According to ITSP and IFTSP Models

In this paper, considering the changes in the overall functional area of the Boluo Lake National Nature Reserve, the functional area of each lake bubble was adjusted. According to Table 6, after model optimisation, the fish pond area in the Toudaogang Reservoir changed by [−0.72, −0.32] × 10^4^ hm^2^ compared to that determined using the ITSP model, and the crab pond area changed by [0.00, −0.68] × 10^4^ hm^2^. Compared to the value obtained using the ITSP model, the fish pond area of Boluo Lake changed by [3.68, −1.04] × 10^4^ hm^2^. The upper limit of the crab pond area and the lower limit of the reed wetland increased by 0.44 × 10^4^ hm^2^ and 1.58 × 10^4^ hm^2^, respectively, and the upper limit of the marsh wetland decreased by 0.66 × 10^4^ hm^2^. Compared to the value obtained using the ITSP model, the upper and lower limits of the Mobopao fish pond area increased by [0.77, 0.33] × 10^4^ hm^2^, respectively, and the functional areas of other regions did not change. Compared to the values obtained using the ITSP model, the upper limit of the crab pond area increased by 0.23 × 10^4^ hm^2^ in Yuanbaowapao, the lower limit of the marsh wetland area increased by [0.48, 0.18] × 10^4^ hm^2^, and the sizes of the other functional areas did not change. Compared to the values obtained using the ITSP model, the upper limit of the fish pond area and reed wetland area decreased by 2.68 × 10^4^ hm^2^, the upper and lower limits of the reed wetland changed by [−0.48, −1.10] × 10^4^ hm^2^, and the sizes of the other functional areas remained unchanged. Overall, the upper and lower limits of the total functional area in the Boluo Lake National Nature Reserve as simulated using the proposed model are [13.44, 23.32] × 10^4^ hm^2^, respectively. Moreover, the lower limit is 5.31 × 10^4^ hm^2^ higher than that obtained using the ITSP model, and the possibility level is 0.32, indicating that the satisfaction of the decision-makers is low, but it is less difficult for the functional areas to return to the lower limit of the simulation result interval used in the IFTSP model. The upper limit is 5.30 × 10^4^ hm^2^ lower than that determined using the ITSP model, and the possibility level is 0.64, indicating that the satisfaction of decision-makers is high, but it is difficult for the functional area of the region to return to the upper limit of the simulation result interval used in the IFTSP model, and the size of the functional area is 51.78% smaller. After introduction of the fuzzy method, the width of the restoration function area of the IFTSP model decreased compared to that determined using the ITSP model, thus reducing the upper limit and increasing the lower limit. At the same time, it reduces the risk that the actual recovery function area cannot reach the upper limit, thereby mitigating the risks involved in decision-making.

### 4.4. Comprehensive Comparative Analysis for Boluo Lake Wetland Recharge Project by Using IFTSP and ITSP Models

In this paper, IFTSP method is first applied to the ecological water replenishment project planning of Boluo Lake. The IFTSP model constructed takes into account that the planned area of the lake bubble in the Boluo Lake National Nature Reserve is an interval value. When ecological replenishment of the Boluo Lake National Nature Reserve is implemented, the restoration of the maximum area of the lake bubble can lead to wastage of water resources. If the restored area is considerably small, it can affect the habitat and reproduction of aquatic organisms and migratory birds in the reserve, which have fuzzy and uncertain characteristics. Therefore, the planned area of the lake bubble is defined as a fuzzy parameter when constructing the model. Compared with ITSP model, the objective function of IFTSP model constructed in this paper is fuzzy membership degree as shown in constraint 6, so as to shorten the area of lake bubble function area. Moreover, the minimum constraint of ecological replenishment is added as shown in constraint 7, the ecological replenishment obtained by the interval two-stage stochastic programming model was blurred. At the same time, constraints 10, 16 and 18 are introduced to blur functional area and lake bubble area, so as to scientifically and rationally allocate water resources. This paper makes a comparative analysis from three perspectives: ecological water replenishment, flood diversion, and restoration area. After introduction of the fuzzy mathematical programming method for optimisation, the model simulation indicates that the ecological water replenishment decision interval range of the reserve is reduced by 58.17%. The functional area division of the Boluo Lake National Nature Reserve is adjusted, sizes of several functional areas are changed (for example, the area covered by fish ponds in Boluo Lake is reduced by 1.07 × 10^4^ hm^2^, and the area covered by fish ponds in Mobopao is increased by 1.04 × 10^4^ hm^2^), and overall functional area is reduced by 51.78%. The range of engineering flood diversion intervals for the five lake bubbles in the three water years of low flow, normal flow, and high flow is significantly reduced (the overall flood diversion volume of the Toudaogang Reservoir has the highest reduction range of 88.09% in the three scenarios). In summary, the IFTSP model constructed in this paper avoids the waste of water resources, shortens the optimal decision-making interval, and improves the decision-making efficiency of decision-makers.

### 4.5. Evaluation of Practicability of IFTSP Model

The IFTSP model constructed herein fuzzified the functional area of the lake bubble to better address the problem of unclear delineation of protected areas. The model results are in the form of intervals, and the interval range is significantly shorter than that of the ITSP model, which improves the decision-making efficiency of decision-makers and facilitates them to recharge Boluo Lake according to the actual situation. The IFTSP model is not only suitable for the Boluo Lake National Nature Reserve but can also be used as a theoretical guide for water replenishment projects in other parts of China and in other countries worldwide. When implementing a water replenishment project for a lake bubble, the general goal is to restore the water level or replenish water. However, owing to the influence of natural factors, such as river erosion and rainwater erosion, the cross section of the lake bubble is not a regular rectangle and is often in a tortuous and irregular state. Therefore, when implementing water replenishment projects, to reach the target water level height range in lake bubbles, the lake bubble area is considered as an interval. Considering the uncertainty of the lake bubble area, it is set as a fuzzy parameter in ecological water replenishment. Meanwhile, different regions have different climates, and in areas with changeable climate, precipitation can be divided into five scenarios: extremely low flow year, low flow year, normal flow year, high flow year, and extremely high flow year. In the ecological water replenishment project of Boluo Lake National Nature Reserve described herein, the influence of human activities on the water replenishment project was negligible, but when implementing water replenishment projects in densely populated areas, it is necessary to consider the impact of human activities, such as industrial water and farming activities, on wetland water replenishment. Therefore, the application of the proposed model should be adjusted according to the influence of local factors such as climate, topography, and anthropogenic activities, and the constraints should be selected according to the actual local conditions to construct an IFTSP model that meets the local reality. This can help formulate an ecological water replenishment scheme that is conducive to the coordinated development of local economic and ecological functions.

## 5. Conclusions

Herein, we developed an IFTSP model for wetland ecological water replenishment optimisation in the Boluo Lake National Nature Reserve. The model fully considered the fuzzy uncertainty of the water replenishment area in lake bubbles and optimised water replenishment allocation for wetlands based on the minimum water resource consumption criterion. After simulation and optimisation using the proposed model, the decision-making ranges of ecological water replenishment in the lake bubble area decreased by 58.17%, which ensured the stability of the ecosystem, shortened the decision-making space, and improved the decision-making efficiency of decision-makers. The range of flood diversion volume decreased significantly, and the proportion of interval width reduction in the three scenarios was greater than 57.00%, which ensured full utilisation of the flood resources and avoided the wastage of water resources. In addition, after introduction of the fuzzy programming method, the total functional area of the restoration reserve decreased by 51.78%, and the upper limit of the restoration area increased significantly. Meanwhile, the system failure risk that the upper limit value cannot be obtained after model optimisation decreased. The result of the interval fuzzy two-stage model shortened the scope of decision-making and reduced the risk of decision-making, thus facilitating managers to make better decisions. Therefore, the interval fuzzy two-stage stochastic programming model developed herein fully considers the uncertainty of regional area, allocates water resources scientifically and rationally, and restores the ecological function and stability of wetland systems. It can help decision-makers to solve the problem of ecological water replenishment in wetlands in a better manner.

## Figures and Tables

**Figure 1 ijerph-19-05218-f001:**
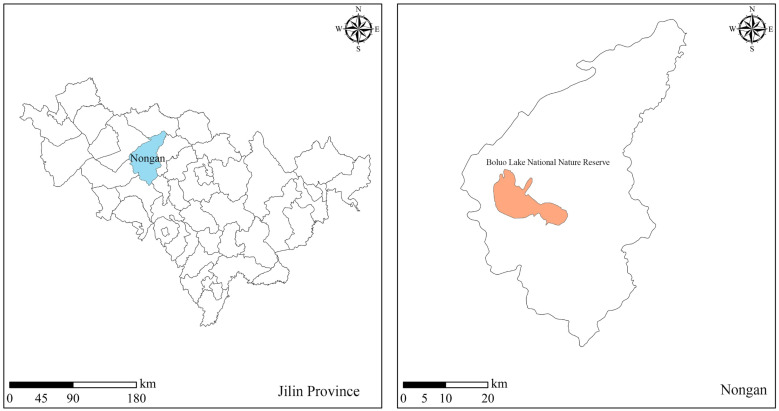
Geographical location of study area.

**Figure 2 ijerph-19-05218-f002:**
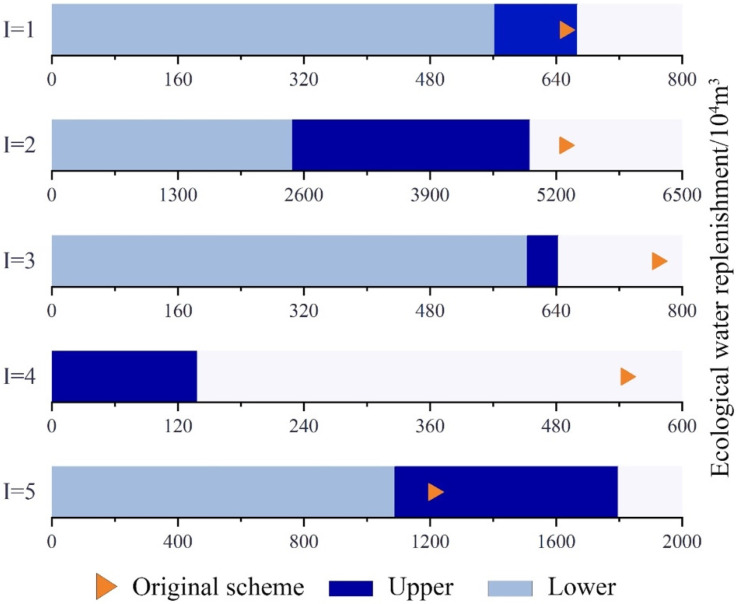
Comparison between recommended project scheme and the IFTSP-based model optimisation scheme for ecological water replenishment of each lake in the Boluo Lake Wetland National Nature Reserve. (I = 1 to 5 represent the Toudaogang Reservoir, Boluo Lake, Mobopao, Yuanbaowapao, and Aobaotupao, respectively).

**Figure 3 ijerph-19-05218-f003:**
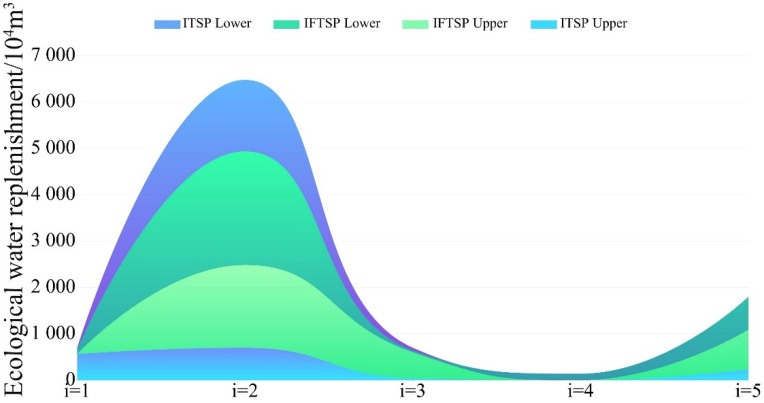
Comparison between IFTSP model and ITSP model optimisation schemes for the ecological water replenishment project of each lake in Boluo Lake National Nature Reserve. (i = 1 to 5 represent the Toudaogang Reservoir, Boluo Lake, Mobopao, Yuanbaowapao, and Aobaotupao, respectively).

**Table 1 ijerph-19-05218-t001:** Ecological replenishment water for each lake bubble in Boluo Lake National Nature Reserve.

Lake Bubble	Total Water Replenishment (×10^4^ m^3^)		
	The Recommended Scheme for Project	IFTSP Model	Range of Variation
Toudaogang Reservoir	651.46	[562.40, 667.49]	[−13.67%, 2.46%]
Boluo Lake	5289.64	[2483.99, 4934.56]	[−53.04%, −6.71%]
Mobopao	768.72	[603.81, 643.41]	[−21.45%, −16.30%]
Yuanbaowapao	546.50	[0.00, 138.90]	[−100.00%, −74.58%]
Aobaotupao	1211.64	[1089.66, 1798.29]	[−10.07%, 48.42%]

**Table 2 ijerph-19-05218-t002:** Comparison between IFTSP model and ITSP model optimisation schemes for ecological water replenishment of each lake bubble in Boluo Lake National Nature Reserve.

Name of Lake Bubble	Total Water Replenishment (×10^4^ m^3^)		
	ITSP Model	IFTSP Model	Reduced Scale of Decision Space
Toudaogang Reservoir	[562.40, 692.94]	[562.40, 667.49]	19.50%
Boluo Lake	[697.50, 6476.21]	[2483.99, 4934.56]	57.59%
Mobopao	[41.60, 698.60]	[603.81, 643.41]	93.97%
Yuanbaowapao	[0.00, 138.90]	[0.00, 138.90]	0.00%
Aobaotupao	[234.40, 1798.29]	[1089.66, 1798.29]	54.69%

**Table 3 ijerph-19-05218-t003:** Flood diversion volumes of each of the lake bubbles in the Boluo Lake under different scenarios.

Lake Bubble	Flood Diversion Volume (×10^4^ m^3^)			
	The Recommended Scheme for Project	IFTSP Model		
		*h* = 1	*h* = 2	*h* = 3
Toudaogang Reservoir	0.00	[270.36, 335.63]	[84.54, 177.12]	[84.54, 177.12]
Boluo Lake	621.00	[0.00, 0.00]	[0.00, 0.00]	[0.00, 0.00]
Mobopao	295.00	[269.40, 307.21]	[269.40, 478.80]	[12.60, 12.60]
Yuanbaowapao	2.00	[206.55, 289.80]	[56.70, 289.80]	[56.70, 56.70]
Aobaotupao	14.00	[227.37, 556.39]	[227.37, 556.39]	[227.37, 556.39]
Total	932.00	[973.68, 1489.03]	[638.01, 1502.11]	[381.21, 802.81]

**Table 4 ijerph-19-05218-t004:** Comparison between IFTSP and ITSP model optimisation schemes for flood diversion volumes of each of the lake bubbles in Boluo Lake under different scenarios.

Lake Bubble	The Recommended Scheme for Project	ITSP Model			IFTSP Model		
		*h* = 1	*h* = 2	*h* = 3	*h* = 1	*h* = 2	*h* = 3
Toudaogang Reservoir	0.00	[0.00, 335.63]	[0.00, 335.63]	[0.00, 335.63]	[270.36, 335.63]	[84.54, 177.12]	[84.54, 177.12]
Boluo Lake	621.00	[0.00, 0.00]	[0.00, 0.00]	[0.00, 0.00]	[0.00, 0.00]	[0.00, 0.00]	[0.00, 0.00]
Mobopao	295.00	[0.00, 431.10]	[0.00, 478.80]	[0.00, 12.60]	[269.40, 307.21]	[269.40, 478.80]	[12.60, 12.60]
Yuanbaowapao	2.00	[0.00, 289.80]	[0.00, 289.80]	[0.00, 56.70]	[206.55, 289.80]	[56.70, 289.80]	[56.70, 56.70]
Aobaotupao	14.00	[0.00, 910.70]	[0.00, 910.70]	[0.00, 910.70]	[227.37, 556.39]	[227.37, 556.39]	[227.37, 556.39]
Total	932.00	[0.00, 1967.23]	[0.00, 2014.93]	[0.00, 1315.63]	[973.68, 1489.03]	[638.01, 1502.11]	[381.21, 802.81]

**Table 5 ijerph-19-05218-t005:** Functional area of each land type in different study areas within Boluo Lake.

Lake Bubble	Functional Area	Area of Functional Areas (×10^4^ hm^2^)		
		Original Scheme	The Recommended Scheme for Project	IFTSP Model
Toudaogang Reservoir	Fish pond	[0.03, 0.05]	0.40	[0.00, 0.72]
	Crab pond	[0.00, 0.00]	0.00	[0.00, 0.00]
	Reed wetland	[0.06, 0.12]	1.00	[0.00, 0.96]
	Marsh wetland	[0.02, 0.04]	0.34	[0.00, 0.33]
Boluo Lake	Fish pond	[0.69, 1.30]	10.84	[6.62, 6.96]
	Crab pond	[0.00, 0.00]	0.00	[0.00, 0.44]
	Reed wetland	[0.26, 0.48]	4.00	[3.65, 3.84]
	Marsh wetland	[0.13, 0.24]	2.00	[1.92, 2.11]
Mobopao	Fish pond	[0.07, 0.13]	0.00	[0.77, 1.04]
	Crab pond	[0.00, 0.00]	1.07	[0.00, 0.00]
	Reed wetland	[0.03, 0.06]	0.50	[0.00, 0.48]
	Marsh wetland	[0.06, 0.12]	1.00	[0.00, 0.96]
Yuanbaowapao	Fish pond	[0.00, 0.00]	0.00	[0.00, 0.00]
	Crab pond	[0.00, 0.00]	0.00	[0.00, 1.37]
	Reed wetland	[0.01, 0.01]	0.10	[0.00, 0.10]
	Marsh wetland	[0.03, 0.06]	0.50	[0.48, 0.66]
Aobaotupao	Fish pond	[0.00, 0.00]	0.00	[0.00, 0.00]
	Crab pond	[0.20, 0.37]	3.05	[0.00, 2.14]
	Reed wetland	[0.03, 0.06]	0.50	[0.00, 0.48]
	Marsh wetland	[0.05, 0.09]	0.76	[0.00, 0.73]

**Table 6 ijerph-19-05218-t006:** Comparison between IFTSP and ITSP model optimisation schemes for functional area of each land type in different study areas within Boluo Lake.

Lake Bubble	Functional Area	Area of Functional Areas (×10^4^ hm^2^)		
		ITSP Model	IFTSP Model	Range of Variation
Toudaogang Reservoir	Fish pond	[0.72, 1.04]	[0.00, 0.72]	[−0.72, −0.32]
	Crab pond	[0.00, 0.68]	[0.00, 0.00]	[0.00, −0.68]
	Reed wetland	[0.00, 0.96]	[0.00, 0.96]	[0.00, 0.00]
	Marsh wetland	[0.00, 0.33]	[0.00, 0.33]	[0.00, 0.00]
Boluo Lake	Fish pond	[2.94, 8.00]	[6.62, 6.96]	[3.68, −1.04]
	Crab pond	[0.00, 0.00]	[0.00, 0.44]	[0.00, 0.44]
	Reed wetland	[2.07, 3.84]	[3.65, 3.84]	[1.58, 0.00]
	Marsh wetland	[1.92, 2.77]	[1.92, 2.11]	[0.00, −0.66]
Mobopao	Fish pond	[0.00, 0.71]	[0.77, 1.04]	[0.77, 0.33]
	Crab pond	[0.00, 0.00]	[0.00, 0.00]	[0.00, 0.00]
	Reed wetland	[0.00, 0.48]	[0.00, 0.48]	[0.00, 0.00]
	Marsh wetland	[0.00, 0.96]	[0.00, 0.96]	[0.00, 0.00]
Yuanbaowapao	Fish pond	[0.00, 0.00]	[0.00, 0.00]	[0.00, 0.00]
	Crab pond	[0.00, 1.14]	[0.00, 1.37]	[0.00, 0.23]
	Reed wetland	[0.00, 0.10]	[0.00, 0.10]	[0.00, 0.00]
	Marsh wetland	[0.00, 0.48]	[0.48, 0.66]	[0.48, 0.18]
Aobaotupao	Fish pond	[0.00, 2.68]	[0.00, 0.00]	[0.00, −2.68]
	Crab pond	[0.00, 2.14]	[0.00, 2.14]	[0.00, 0.00]
	Reed wetland	[0.48, 1.58]	[0.00, 0.48]	[−0.48, −1.10]
	Marsh wetland	[0.00, 0.73]	[0.00, 0.73]	[0.00, 0.00]

## Data Availability

The data presented in this study are available contained within the article.

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
