# Peer review of "Optimization of Ecological Water Replenishment Scheme Based on the Interval Fuzzy Two-Stage Stochastic Programming Method: Boluo Lake National Nature Reserve, Jilin Province, China"

_ijerph, 2022, doi:10.3390/ijerph19095218_

Round 1
Reviewer 1 Report
Thanks to the authors for the detailed description. We understand the opportunity to extend the previous study under the given conditions, but it is ethics for this paper to be presented as part II of the previous one. In conclusion: 1. it can be successfully published in the same magazine as the previous one; 2. only those original results will be presented, compared to what was previously published; 3. it is easier to make an analysis compared to the results of the first study. 4. It is certain that it is not possible to write a second article in which more than 50% of the information is the same, but presented in other words. We encourage authors to republish this article in the same journal as the first part.Author Response
Please see the attachment.

Reviewer 2 Report
The article seems to be well-structured; the authors carried out a deepened analyses and provide numerous elements to confirm their conclusions. However, some additions are, in my opinion, mandatory, to make the present study suitable for publication.
The abstract must be completely rewritten; it should be more concise and it should directly explain in what the author's proposal consisted of and also report the main results achieved.
The meaning of all the symbols used innequations 1 - 5 must be clearly explained below those equations.
In section 3, the authors described the model then adopted in the following sections and inserted some references to validate it. I would know which is the author contribution in the realization of this model or, better, which is the contribution to the model produced in this work.
The comparison between the present method with the model currently used must be deepened in the text. I suggest to authors to insert an appropriate sub-section in the Results and Discussion, completely involved on this and to provide more information in this sense, which are needed to well evaluate the novelty introduced with this work.
Round 2
Reviewer 1 Report
Although, in the explanatory letter, the authors present the motivation for which they reanalyzed the research method from the previous article (‘Optimization Model of the Ecological Water Replenishment Scheme for Boluo Lake National Nature Reserve Based on Interval Two-Stage Stochastic Programming’.), in this article they do not refer at all to this fact. As already mentioned, this article cannot be presented separately from the previous one, which is why I suggested submitting it to the same magazine.
I repeat the above, from an ethical point of view, this article cannot be presented as an original work, if it is based on another work, even if the latter had vulnerabilities.
However, if the authors really want such an approach, they must modify the study presentation strategy. Thus, to start from the results obtained in the previous article and explain the vulnerabilities they discovered and the ways in which the new research was conducted to improve the results of the model.
Reviewer 2 Report
The authors correctly answered to all of my comments. Now the article can be accepted, as it is.
Author Response
Thank you for your affirmation of our article, as well as for the publication of our article to provide valuable comments and suggestions, we wish you a happy life.